# Diabetes-Related Dietary Patterns and Endometrial Cancer Risk and Survival in the European Prospective Investigation into Cancer and Nutrition Study

**DOI:** 10.3390/nu17101645

**Published:** 2025-05-12

**Authors:** Luisa Torres-Laiton, Leila Luján-Barroso, Núria Nadal-Zaragoza, Carlota Castro-Espin, Paula Jakszyn, Camilla Panico, Charlotte Le Cornet, Christina C. Dahm, Dafina Petrova, Daniel Ángel Rodríguez-Palacios, Franziska Jannasch, Giovanna Masala, Laure Dossus, Lisa Padroni, Marcela Guevara, Matthias B. Schulze, Renée T. Fortner, Rosario Tumino, Marta Crous-Bou

**Affiliations:** 1Unit of Nutrition and Cancer, Catalan Institute of Oncology–Bellvitge Biomedical Research Institute (ICO-IDIBELL), L’Hospitalet de Llobregat, 08908 Barcelona, Spain; ltorresl@idibell.cat (L.T.-L.);; 2Department of Nutrition, Food Science and Gastronomy, Faculty of Pharmacy and Food Sciences, University of Barcelona, 08028 Barcelona, Spain; 3Department of Public Health, Mental Health and Maternal and Child Health Nursing, Faculty of Nursing, University of Barcelona, 08007 Barcelona, Spain; 4International Agency for Research of Cancer, 69366 Lyon, France; 5Blanquerna School of Health Sciences, Ramon Llull University, 08025 Barcelona, Spain; 6Department of Imaging and Radiotherapy, Fondazione Policlinico Gemelli, 00168 Rome, Italy; 7Division of Cancer Epidemiology, German Cancer Research Center (DKFZ), 69120 Heidelberg, Germany; 8Department of Public Health, Aarhus University, 8000 Aarhus, Denmark; 9Escuela Andaluza de Salud Pública (EASP), 18011 Granada, Spain; 10Instituto de Investigación Biosanitaria ibs. GRANADA, 18012 Granada, Spain; 11Medical Oncology, Hospital Universitario Virgen de las Nieves, 18014 Granada, Spain; 12CIBER in Epidemiology and Public Health (CIBERESP), 28029 Madrid, Spain; 13Department of Epidemiology, Regional Health Council, IMIB-Arrixaca, 30120 Murcia, Spain; 14Department of Molecular Epidemiology, German Institute of Human Nutrition Potsdam-Rehbruecke, 14558 Nuthetal, Germany; 15Clinical Epidemiology Unit, Institute for Cancer Research, Prevention and Clinical Network (ISPRO), 50141 Florence, Italy; 16Department of Clinical and Biological Science, University of Turin, 10124 Torino, Italy; 17Instituto de Salud Pública y Laboral de Navarra, 31003 Pamplona, Spain; 18Navarra Institute for Health Research (IdiSNA), 31008 Pamplona, Spain; 19Institute of Nutritional Science, University of Potsdam, 14558 Nuthetal, Germany; 20Department of Research, Cancer Registry of Norway, Norwegian Institute of Public Health, 0456 Oslo, Norway; 21Hyblean Association for Epidemiology Research, AIRE–ONLUS, 97100 Ragusa, Italy; 22Department of Epidemiology, Harvard T.H. Chan School of Public Health, Boston, MA 02115, USA

**Keywords:** diabetes, dietary patterns, endometrial cancer, etiology, risk factors, survival

## Abstract

Background/Objectives: Endometrial cancer (EC)’s major risk factors include obesity and diabetes, both strongly related with lifestyle choices and dietary factors. Our study aimed to evaluate the relationship between diabetes-related dietary patterns, EC risk, and survival in a population of middle-aged European women. Methods: A total of 285,418 female participants from the European Prospective Investigation into Cancer and Nutrition (EPIC) study were included in the analysis. After a mean time of 10.6 years of follow-up, 1955 incident EC cases were registered; of those, 133 women died from EC. The Empirical Dietary Index for Insulin Resistance (EDIR), the Empirical Dietary Index for Hyperinsulinemia (EDIH), and the Diabetes Risk Reduction Diet (DRRD), were estimated from dietary information collected at baseline from EPIC participants. Cox proportional hazards regression models were used to evaluate the association between the dietary patterns and EC risk, using hazard ratios (HR), 95% confidence intervals (CI), and adjusting for relevant confounders. Cox and Fine–Gray models were used to assess the association with overall and EC-specific mortality, respectively. Results: Higher adherence to EDIR was associated with an increased risk of EC, multivariable HR for T3vsT1 were 1.17 (95% CI = 1.04 to1.31). However, when BMI was included in the models, these associations became weaker and no longer statistically significant. No associations were observed in relation to adherence to EDIH, DRRD, and EC risk. No associations were found in relation to diabetes-related dietary patterns and mortality. Conclusions: This study highlights the potential role of diabetes related dietary patterns and EC etiology and prevention. Further studies are warranted to better understand the role of etiology-derived dietary patterns and disease prevention and prognosis.

## 1. Introduction

Understanding the relationship between obesity, type 2 diabetes, and cancer requires special attention, as the first two are key lifestyle factors linked to an increased risk of specific types of cancer [1,2]. Endometrial cancer (EC) is the sixth most common cancer in women worldwide, with an incidence of 420,368 cases in 2022; it is expected that by the year 2045, the number will rise to 564,070 [3]. Although it has a favorable prognosis with a 95% five-year relative survival rate in a localized stage [4], in the year 2022, it was responsible for the deaths of 97,723 women [3].

EC may be classified into two types: type I or endometrioid type, which is a hormone-related cancer caused by the gradual buildup of estrogen in the endometrium without the counterbalancing effects of progesterone; and type II, which includes non-endometrioid cancers that are not directly associated with endocrine dysregulation [5]. The main relevant risk factors underlying EC appear to be unopposed estrogen replacement therapy, early menarche, late menopause, nulliparity, diabetes mellitus, and obesity [6]. These last two risk factors are associated with insulin resistance and hyperinsulinemia [5,7], both of which can be influenced by diet. In women with obesity and type 2 diabetes, insulin and leptin levels are elevated, creating a hormonal environment that can promote cancer cell growth [8] by promoting the activation of insulin-like growth factor 1 (IGF-1) [2]. Moreover, insulin and insulin-like growth factors accelerate cell division while inhibiting apoptosis [9], thus creating favorable hormonal conditions for cancer development.

Further, diet plays a crucial role in the risk and survival of various types of cancer [8]. Although a direct relationship between diet and EC remains unclear, probable evidence has been found between coffee consumption as a protective factor and glycemic load (GL) as a risk factor [10]. However, since individuals consume food groups rather than isolated foods, a useful way to examine the relationship between diet and cancer development from an epidemiological perspective is through the study of dietary patterns [11]. The association between diabetes-related dietary patterns and EC risk and mortality has been evaluated in some studies, yielding inconsistent results. Some authors have reported an association between EC risk and diabetes-related patterns mediated by adiposity [12,13], while others have found a reduced EC risk when following dietary patterns protective against diabetes [14], and others have found no significant associations [9,15,16]. Despite this, there is limited evidence on how diet influences the development, treatment, and progression of the disease. Thus, conducting an analysis that focuses on dietary patterns based on underlying pathways involved in the EC etiology and progression is of high interest. Based on this, we aimed to evaluate the relationship between three dietary patterns associated with diabetes and their impact on EC risk and survival in a large prospective cohort study, the European Prospective Investigation into Cancer and Nutrition (EPIC). The Empirical Dietary Index for Hyperinsulinemia (EDIH) and the Empirical Dietary Index for Insulin Resistance (EDIR), which are linked to an increased risk of diabetes, and the Diabetes Risk Reduction Diet (DRRD), which is associated with diabetes prevention. These patterns were specifically chosen because they assess dietary influences on insulin resistance and hyperinsulinemia, both of which play a key role in the development of EC.

## 2. Materials and Methods

### 2.1. Study Population

EPIC is a prospective cohort study conducted between 1992 and 2000, encompassing over 500,000 middle-aged adults. The details on the study have been previously described in detail [17]. At the beginning of the study, participants filled out questionnaires regarding their diet, lifestyle, and medical history, and anthropometric measurements were taken along with blood samples. While some self-reporting was involved, most anthropometric measurements were conducted using standardized protocols across the majority of EPIC centers. M Haftenberger et al. [18] describe, in detail, the protocol for anthropometric measurements. All participants provided written informed consent during recruitment. Lifestyle factors such as tobacco smoking, alcohol intake, and physical activity were assessed, with physical activity being self-reported by participants using a set of standardized questions across countries. Additionally, information on menstrual and reproductive history, contraceptive methods, menopausal status, and use of hormone therapy was collected [17]. In our sample, we excluded male participants, women with incomplete lifestyle or dietary data, and those with implausible daily consumption values. Finally, the current study included 285,418 women from nine countries (Denmark, France, Germany, Italy, The Netherlands, Norway, Spain, Sweden, UK). Incident cancer case registrations were conducted based on population cancer and pathology registries, health insurance records and/or on active follow-up [17]; likewise, death records were obtained through registries and death record collections. Cancer cases were classified according to the International Statistical Classification of Diseases and Related Health Problems. For the present analysis, we identified a total of 1955 cases of EC. After a mean of 10.6 years of follow-up, 380 cases died, of which 133 were due to EC.

### 2.2. Dietary Information

To gather dietary information, EPIC centers primarily used standardized and validated food frequency questionnaires (FFQs)—which included between 88 and 266 food items—and, less frequently, diet history questionnaires [17]. FFQs were either self-administered or conducted face-to-face with interviewers in some EPIC centers. Energy and nutrient intakes were estimated using country-specific food composition tables [17]. In our analysis, three diabetes-related dietary patterns [19,20] were calculated for each participant based on the information available in the literature, and the FFQ data collected at the time of recruitment. EDIH and EDIR are hypothesis-driven dietary patterns [19], while DRRD is an a priori dietary pattern based on a predefined set of criteria [20]. The calculation of all dietary patterns was performed with each food group standardized by 2000 kcal.

The EDIH and EDIR patters were developed by Tabung FK et al. [19], and are based on the calculation of daily intakes of 18 food groups. Although both patterns include the same number of components, the food groups are not exactly identical. Detailed methods for calculating EDIH and EDIR have been described elsewhere [19]. Briefly, food groups were selected based on information from EPIC’s FFQ. The intake of each component was described using mean and standard deviation (SD) values. Z-scores were then calculated by subtracting the mean from each intake value and dividing by the SD. Each z-score was multiplied by its corresponding insulinemic weight, and the resulting values were summed to obtain the final score. The weights we used for the dietary patterns calculations were applied as reported in the original reference [19] (see Appendix A).

EDIH provides a cohort-dependent range from minor to major, where a higher score indicates that the individual consume a diet that may increase levels of hyperinsulinemia—related to the *C*-peptide concentration [21]—while a lower score suggests a potential normo-insulinemic diet. Although the index includes 13 foods positively associated with hyperinsulinemia, our analysis included only 12, as the contribution of French fries was omitted, since the dietary questionnaires used in the EPIC populations did not categorize the consumption of this particular food (Appendix A). On the other hand, a higher EDIR score indicates a diet associated with a greater likelihood of insulin resistance, whereas a lower score suggests a higher degree of insulin sensitivity. In this case, insulin resistance was assessed by the original authors [19] using the ratio between fasting triglycerides and fasting HDL cholesterol, as this approach helps identify seemingly healthy patients that may have insulin resistance [22].

The DRRD was originally developed by Rhee et al. [23], incorporating food components associated with the risk of type 2 diabetes. It was later modified by Kang et al. [20], who classified total fruit intake as a protective factor, while grouping fruit juices with sugar-sweetened beverages as an adverse factor. In addition to food groups and individual foods, it also includes nutrients and the glycemic index. For the derivation of the DRRD, each participant is assigned to quintiles based on their intake of nine different components. These components are then rated on a scale from 1 to 5, reflecting their association with the risk or prevention of type 2 diabetes (Appendix A). The final score ranges from 9 to 45, where a higher score indicates a healthier diet, associated with the prevention of type 2 diabetes [20].

### 2.3. Statistical Analysis

Cox proportional hazards regression models were employed to calculate hazard ratios (HR) and 95% confidence intervals (CI) to prospectively analyze the association between the three dietary patterns and the risk of developing EC [24]. In the risk models, the time of cohort entry was determined by the age of the participants at recruitment, while the time of exit was defined by the age at EC diagnosis, death, end of follow-up, or the last known contact with the participant, whichever occurred first. All risk models were stratified by country and age (by 10-year categories) at recruitment. The dietary pattern scores were correlated using Pearson correlation coefficients as follows: DRRD and EDIR (−0.34), DRRD and EDIH (−0.24), and finally EDIR and EDIH (0.85).

Three multivariable risk models were evaluated. The inclusion of variables was determined based on their relationship with EC risk and survival, plus the results of the Chi-square test, which compared the deviance between models and assessed the model’s fit when adding or removing specific variables. Therefore, it was decided to include the variables that significantly improved the model according to the test. The first model included menopausal status (premenopausal, postmenopausal)—perimenopausal participants were excluded, and postmenopausal includes those with natural menopause and those who had undergone surgical bilateral ovariectomy—smoking status (never smoker, former, active smoker), and the use of hormonal treatment for menopause (no, yes). The second model additionally included BMI as a continuous variable. We performed further analysis by subgroups, including BMI (normal weight < 25 kg/m^2^, overweight and obese ≥ 25 kg/m^2^), menopausal status (premenopausal, postmenopausal), diabetes status (diabetic, non-diabetic), physical activity (active, inactive), and smoking status (never smoker, former, active smoker). A sensitivity analysis for the risk of type I EC was conducted. Analyses for type II EC cases were not performed due to the small sample size (n = 103). An additional mediation analysis was conducted to assess the extent to which the effect on EC risk was mediated by BMI. This was performed based on the difference method [25], a statistical approach that in this case compares estimates from models with and without the BMI as a potential mediator.

To evaluate the relationship between dietary patterns at recruitment and overall and specific EC mortality, Fine–Gray competing risks models were employed [26]. The models accounted for time of entry as age at EC diagnosis, and exit time defined as death or end of follow-up. The mortality models were stratified by country and 10-year categories at diagnosis, and adjusted for potential confounders including tumor type (Type I, Type II), tumor stage (in situ or localized, metastatic, and unknown), BMI (continuous variable), and menopausal status (premenopausal, postmenopausal). Other variables did not contribute significantly according to the analysis of deviance. Subgroup analyses were also conducted, including BMI, diabetes mellitus, menopausal status, physical activity, and smoking status, categorized as previously described.

The three dietary patterns were categorized based on the analysis conducted using splines. Since the EDIH did not fit a linear model, the best representation of the data were achieved using tertiles. Conversely, the DRRD and EDIR results showed a closer fit to a linear model; however, they were also categorized into tertiles to facilitate interpretation of the results. To assess trends across tertiles, the scores were treated as continuous variables and included in the model for calculation purposes. The lowest tertile was used as the reference category for all models. The distribution of tertiles in the DRRD is not homogeneous across the three groups, as a significant number of participants have values that align with the cut-off point of the first tertile. All statistical analyses were performed using Rstudio version 4.4.2.

## 3. Results

The present study included a total of 285,418 women, of whom 1955 developed EC during the follow-up period of 10.6 years. Table 1 presents the baseline characteristics of the included population. The mean age of the women at recruitment was 50.13 years (SD 9.8), with a mean BMI of 24.68 kg/m^2^ (SD 4.3). They were mostly never smokers, physically inactive, and postmenopausal. The majority of participants had two children, used oral contraceptives, and did not use postmenopausal hormonal treatments. Additionally, 1.9% of the women had self-reported diabetes mellitus.

The women diagnosed with EC were older at recruitment (mean 54.75 years SD 7.6) than women without EC, and had a mean age of 63.51 years (SD 8.2) at diagnosis. They had higher BMI (mean 26.85 kg/m^2^ SD 5.3), and were also more likely to be non-smokers and physically inactive. A higher proportion of EC cases experienced early menarche and late menopause, and a greater percentage were postmenopausal, were more likely to use postmenopausal hormonal treatment, and had a lower use of oral contraceptives. Finally, a higher proportion of EC cases were self-reported diabetics (3.4%). Details related to adherence to the dietary patterns and the characteristics of the participants by categories of adherence to each dietary pattern are presented in Appendix A.

Figure 1 illustrates the cumulative incidence of EC cases during the follow-up period, for each dietary pattern. Regarding EDIR and EDIH, a slightly higher incidence of EC over time was observed with higher adherence to the pattern. No associations were observed regarding DRRD. Table 2 presents the association between adherence to the diabetes-related dietary patterns and the risk of EC. The multivariable models indicate an increased risk of EC in women with higher adherence to EDIR HR_T3vsT1_ 1.17 (95% CI = 1.04 to 1.31; P_trend_ = 0.008). However, when the models where additionally adjusted for BMI, the associations were no longer statistically significant (HR_T3vsT1_ = 1.03, 95% CI = 0.91 to 1.19; P_trend_ = 0.61). Based on this, we additionally evaluated the proportion of the association between EDIR and EC risk that was mediated by BMI. In this analysis, BMI accounted for 79% (*p* = 0.001) of the observed association. No associations were observed in relation to adherence to the EDIH and DRRD.

In general, no statistically significant heterogeneity was observed when subgroup analyses were performed in relation to diabetes, menopausal status, BMI, physical activity and smoking status (Appendix A).

The main characteristics of the EC cases included in the mortality analysis are shown in Appendix A. Women who died from EC had a higher BMI compared to women with other causes of death. Figure 2 presents the mortality curves in relation to adherence to each dietary pattern. No significant modification of the mortality rate over the follow-up period was observed for any of the three diabetes-related dietary patterns. Results of the association analysis between adherence to the dietary patterns and overall and EC-specific mortality are presented in Table 3. No significant associations were found with either overall or EC-specific mortality. Subgroup analysis, shown in Appendix A, revealed some heterogeneity among BMI subgroups for EDIH and EDIR, as well as among menopausal status subgroups for all three patterns. However, no significant associations were found for physical activity or smoking status in relation to either overall or EC-specific mortality.

## 4. Discussion

This is the most comprehensive study assessing the relationship between dietary patterns related to either diabetes prevention or diabetes-related mechanisms and EC risk and survival. In a large population of middle aged European women, including 1.955 EC cases, we found that higher adherence to a diabetes-related dietary pattern linked with insulin resistance, was associated with an increased risk of EC. These associations attenuated when BMI was accounted for, as the analysis showed that BMI contributed to 79% of the observed relationship. No associations were observed with a diet focused on diabetes prevention. Moreover, none of the evaluated diabetes-related dietary patterns appeared to impact on overall or EC-specific mortality. Our results suggest a complex interplay between diet, obesity, and the risk of EC, in which insulin-related pathways may play an etiological role. The effect of diabetes-related diets on EC risk is not independent; rather, it is largely explained by the relationship between BMI and EC risk.

To our knowledge, only other three studies have explored similar associations. Romanos-Nanclares et al. evaluated the association between EDIH and EC risk in the context of the Nurses’ Health Study [12]. Similarly to our results, before adjustment for BMI, there was a statistically significant association between EDIH and EC risk (HR_Q5vsQ1_ = 1.58, 95% CI = 1.34 to 1.87; P_trend_ = <0.001). However, attenuated associations were also reported when BMI was accounted for in their analyses, with an HR_Q5vsQ1_ = 1.01, 95% CI = 0.85 to 1.21; P_trend_ = 0.92) for EDIH in their sample of 1.462 cases of type I EC [12]. A study involving 112,468 women from the Women’s Health Initiative and 403 EC cases [13] reported similar findings, showing an increased risk of EC associated with adherence to the EDIH pattern before adjusting for BMI, and particularly for those of endometrioid type. However, after adjusting for BMI, the associations lost statistical significance for both overall (HR_Q5vsQ1_ of 1.18, 95% CI = 0.84 to 1.68; P_trend_ = 0.58) and endometrioid type (HR_Q5vsQ1_ of 1.25, 95% CI = 0.82 to 1.91; P_trend_ 0.29). In an Italian case–control study involving 454 cases of EC and 908 controls [14], they found that women with high adherence to the DRRD had a reduced risk of EC (OR = 0.73, 95% CI = 0.55 to 0.97). Nevertheless, several limitations of case–control designs in evaluating dietary-related associations have been previously reported, and limits the ability to compare the results with those of a cohort study, as if even when an association between diet and cancer is observed, it remains plausible that dietary differences could be a result, rather than a cause of the cancer [27].

Other studies have evaluated the relationship between diabetes-related dietary patterns and other cancer outcomes. Greater adherence to EDIH has been associated with an increased risk of liver cancer in postmenopausal women [28], kidney cancer [29], breast cancer [30], and colorectal cancer in women [31,32]. A recent meta-analysis supports these findings, demonstrating that higher adherence to the EDIH pattern is significantly associated with an increased overall cancer incidence, particularly among females, digestive cancers and breast cancer [33]. Conversely to prior references, and in accordance with ours, other authors found that after adjusting the risk models for BMI the previously statistically significant association was attenuated in both EDIR and EDIH in the context of the 142 hepatocellular carcinoma cases of the Nurses’ Health Study and the Health Professionals Follow-up Study [34,35]. Additionally regarding pancreatic cancer risk, adherence to EDIH does not appear to increase the risk per 1 SD increment [36].

Although we did not find a significant relationship between adherence to DRRD and EC, previous research suggests potential benefits of the DRRD, related to liver cancer [37], especially in participants with a higher BMI [38], renal cancer [39], and breast cancer risk, even after adjusting for BMI and weight change since age eighteen [20].

In our risk models, EDIH and EDIR showed positive associations with EC (both having similar effect sizes and in the same direction). However, only EDIR reached statistical significance, despite the high correlation between the two patterns. This difference may be due to EDIR being more effective at capturing variation in chronic metabolic dysfunction [19]. Therefore, EDIR may serve as a more accurate or sensitive indicator of the metabolic processes underlying the associations observed. In the case of the DRRD, a dietary pattern that promotes the intake of multiple healthful components, its association with diabetes reduction may involve multiple metabolic pathways beyond insulin sensitivity. This broad focus could be one possible explanation for the lack of significant findings in our results, as it may dilute the specific effects related to diabetes.

Several studies have investigated the potential role of diabetes-related dietary patterns in cancer mortality with conflicting results. Regarding colon cancer, some studies suggest that adherence to EDIH is not associated with mortality [40]; however, other studies linked EDIH adherence to poorer colon cancer survival [41], and higher overall cancer mortality [33,42]. In a cohort of 13,270 breast cancer cases, DRRD was associated with a lower overall mortality, but not with cancer-specific mortality [43]. Studies that observed significant associations were conducted in cancers with higher incidence rates and, consequently, a greater absolute number of recorded deaths, despite having survival rates similar to those in our study. This likely enhanced their statistical power to detect such effects. In contrast, the relatively small number of EC-specific deaths (n = 133) in our cohort may have limited our statistical power to detect meaningful associations. Other possible explanations for these null findings could be related to the characteristics of the cohort and the timing of data collection. The FFQs were administered at recruitment, on average 8.7 years before diagnosis, and the follow-up period from diagnosis to the end of follow-up—due to EC death or the conclusion of the cohort—lasted about 7.7 years. This results in an average of 16.4 years between recruitment and EC-related deaths. Over this timeframe, no reassessments of dietary intake were conducted, which may have contributed to the lack of observed associations, considering that dietary habits may change during cancer development, either as a direct consequence of the disease itself or as a result of treatment-related side effects. Nonetheless, to the best of our knowledge, this is the first study to comprehensively evaluate the association between diabetes-related dietary patterns, and EC survival. Even though no associations were observed, further research is warranted to better understand the potential impact of such dietary patterns.

The dietary patterns evaluated in this study were selected because they are potentially involved in the underlying etiological mechanisms involved in EC, since type 2 diabetes is associated with insulin resistance and hyperinsulinemia, and these conditions may, in turn, influence endometrial carcinogenesis [1]. Endometrial tissue has various cell types that respond to hormones, including insulin in the bloodstream, through endometrial insulin receptors [5]. Insulin resistance or hyperinsulinemia—often resulting from a higher BMI—can lead to a decrease in the concentration of sex hormone-binding globulin (SHBG) through a negative feedback mechanism [5]. This reduction in SHBG increases the proportion of free estradiol, as SHBG has specific binding sites for estrogens, and most circulating estradiol is normally bound to this protein [44], thereby enhancing unopposed estrogen exposure in the endometrium. Additionally, insulin is a hormone with antiapoptotic activity, and endometrial cancer cell lines appear to express high-affinity insulin receptors [45]. Dysregulation of insulin, such as in the presence of hyperinsulinemia, may lead to the upregulation of the growth hormone receptor (GHR), consequently increasing hepatic production of IGF-I [46]. Since IGF-I exhibits much stronger mitotic and antiapoptotic activity than insulin, this characteristic may contribute to tumor growth and metastasis, ultimately resulting in both metabolic and mitogenic effects [46,47]. Additionally, a relationship between EC risk and increasing serum levels of *C*-peptide, a component that serves in EDIH as the biomarker of hyperinsulinemia, has been described [48].

Similarly, GL has strong evidence linking it to an increased EC risk, according to the WCRF, as a long-term consumption of a high-glycemic-load diet leads to hyperinsulinemia [10], stimulating the previously described mechanism. GL is calculated by multiplying the glycemic index (GI) by the total available grams of carbohydrate in a given amount of food [49]. In the case of GI, the WCRF classifies it as having limited not conclusive evidence regarding EC risk [10]. The DRRD includes GI, but our results did not show a protective effect with this pattern, similar to other studies that have not revealed significant associations [9,16]. However, their relationship must continue to be studied, as these isolated variables alone may not fully reflect total long-term insulin exposure. Sedentary habits understood as sitting time fall into a category of limited suggestive risk as it might be associated with insulin resistance [10]; even so, in our analysis, no significant risk was reported (Appendix A). A similar phenomenon was observed for menopausal status. Although hormonal changes during menopause may influence the risk of developing EC, no statistically significant heterogeneity was found in our subgroup analysis. This lack of significance may be due—both in this and other subgroup analyses—to differences in the number of participants in each category, which may have limited the ability to detect meaningful associations. Even so, the joint evidence of other authors plus our findings support the biological plausibility of hypothesizing that greater adherence to dietary patterns linked to insulin resistance may contribute to the development of EC; however, when relating it to prognosis, the relationship does not seem to be so clear.

Our study is not exempt of limitations. Dietary intake questionnaires were administered only once at baseline, approximately 8.76 years before diagnosis, which does not account for changes in dietary patterns over lifetime or after cancer diagnosis. However, in the study by A. Romanos-Nanclares et al. [12], where FFQs were administered every four years, the results remained largely unchanged whether dietary intake was assessed at baseline, or when more recent dietary assessments were included. Furthermore, in some countries, self-reported questionnaires were used, potentially introducing bias due to memory recall errors or lack of familiarity with standardized food portion sizes among participants. We focused on calculating and analyzing only three dietary patterns related to both risk and protection of diabetes mellitus; however, other patterns related to the biochemical mechanisms of diabetes mellitus could have been evaluated to explore potential relationships with food groups not included in our analysis. Additionally, because of the observational nature of the study, residual confounding is possible, although we controlled for significant confounders.

In contrast, some of the strengths of our study lie in the multifactorial analysis of key indicators related to EC, such as BMI, diabetes mellitus, menopausal status, smoking status and use of hormonal treatment during menopause. We employed standardized dietary patterns that have been validated in other studies directly linked to *C*-peptide production, TAG: HDL cholesterol ratio, and diabetes prevention. Moreover, the FFQs have been validated, and their reproducibility is reliable. Other strengths of this study include its prospective design, the large number of participants, the long follow-up from the date of diagnosis, and detailed information on potential confounders. Furthermore, the dietary components we used to derive dietary patterns effectively represent the main food groups consumed by the European population. Access to data from a large prospective cohort as EPIC, which includes women from multiple countries, ensures a diverse and representative sample. By analyzing dietary patterns, we offer a more comprehensive and global perspective of the dietary characteristics of the population under study, and its synergistic effect, rather than focusing on individual foods or nutrients in isolation.

## 5. Conclusions

Our findings from a large prospective cohort study suggest that higher adherence to a diabetes-related dietary pattern—especially related to insulin resistance—might have an impact on EC risk. No associations have been observed in relation to either overall or cancer-specific mortality. BMI appears to explain this association. Consequently, specific recommendations encouraging women to maintain a healthy body composition through lifestyle changes may help reduce the incidence of EC. The underlying biological mechanisms, as well as the potential impact of nutritional intervention studies, need to be further studied.

## Figures and Tables

**Figure 1 nutrients-17-01645-f001:**
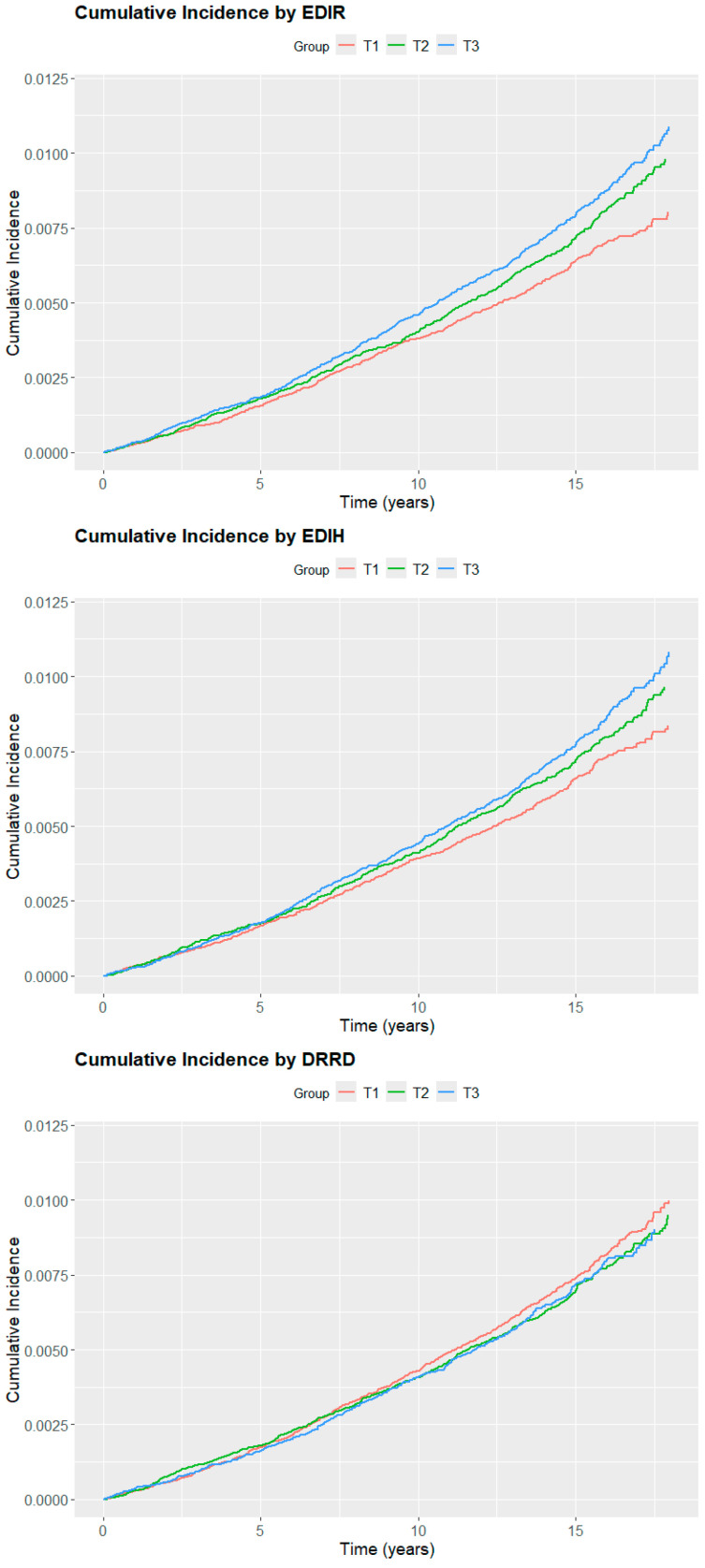
Cumulative incidence (unadjusted) of endometrial cancer during the 10.6 years of follow-up period according to adherence to diabetes-related dietary patterns. T1: first tertile T2: second tertile T3: third tertile.

**Figure 2 nutrients-17-01645-f002:**
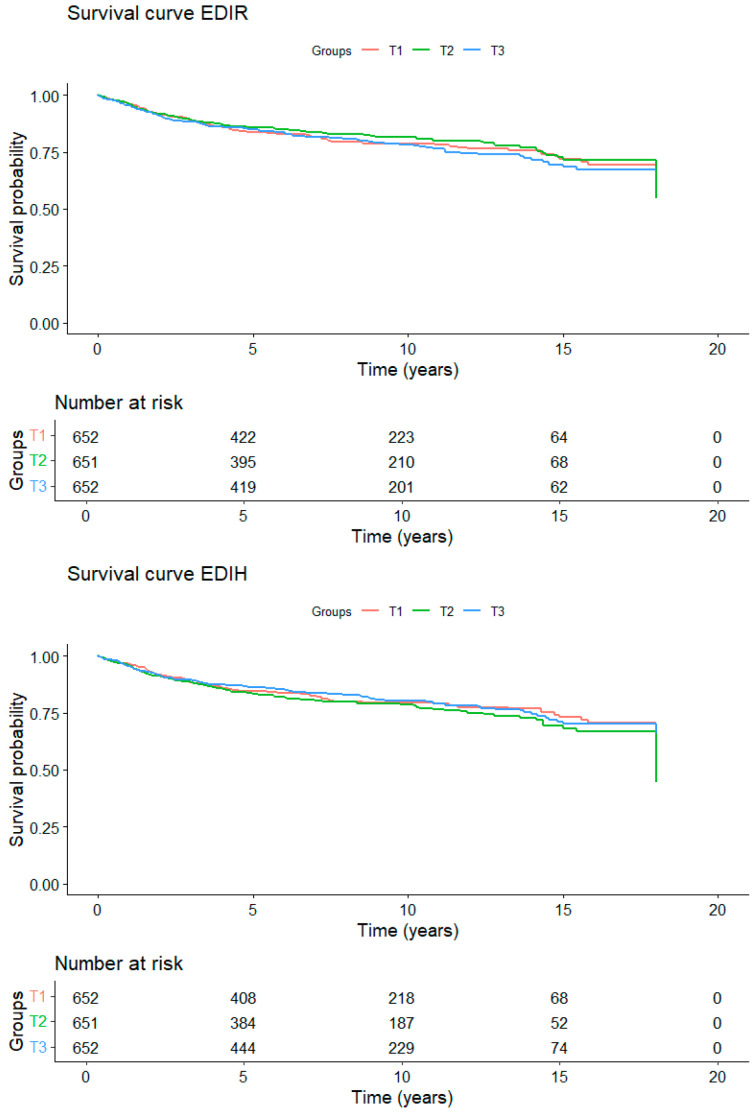
Kaplan–Meier survival curves (overall mortality) over time according to adherence to diabetes-related dietary patterns. Multivariable model stratified by age at diagnosis and country, and adjusted by tumor type (Type I, type II), stage of the tumor (in situ, metastatic, unknown), menopausal status (premenopause, postmenopause) and BMI (continuous).

**Table 1 nutrients-17-01645-t001:** Baseline characteristics of the 285,418 women and the 1955 endometrial cancer (EC) cases in the European Prospective Investigation into Cancer and Nutrition (EPIC) population.

		Participants (n = 285,418)	%	EC Cases (n = 1955)	%
Country	The Netherlands	22,175	7.8	153	7.8
	Spain	22,780	8.0	176	9.0
	Germany	23,303	8.2	98	5.0
	Denmark	24,471	8.6	281	14.4
	Sweden	25,702	9.0	241	12.3
	Italy	27,761	9.7	199	10.2
	Norway	32,416	11.4	222	11.4
	United Kingdom	46,079	16.1	275	14.1
	France	60,731	21.3	310	15.9
Age at recruitment (years)	<40	38,089	13.3	50	2.6
	40 to <50	98,005	32.3	436	22.3
	50 to <60	103,904	36.4	973	49.8
	≥60	45,420	15.9	496	25.4
	mean (SD)	50.13 (9.8)		54.75 (7.6)	
Age at Diagnosis (years)	<50	/	/	84	4.3
	50 to <60	/	/	574	39.4
	60 to <70	/	/	866	44.3
	≥70	/	/	431	22.0
	mean (SD)	/	/	63.51 (8.2)	
Educational level	None	10,097	3.6	98	5.0
	Primary	63,920	22.7	538	27.5
	Technical	62,792	22.0	449	23.0
	Secondary	69,494	24.7	438	22.4
	Longer (University)	68,320	24.3	350	17.9
	Unknown	10,795	3.8	82	4.2
BMI (kg/m^2^)	<18.5	6185	2.2	19	1.0
	18.5 to <25	168,506	59.0	824	42.1
	25 to <30	79,302	27.8	655	33.5
	>30	31,425	11.0	457	23.4
	mean (SD)	24.68 (4.3)		26.85 (5.3)	
Waits circumference (cm)	<88	157,784	55.3	917	46.9
	≥88	40,670	14.2	505	25.8
	Unknown	86,964	30.5	533	27.3
	mean (SD)	79.25 (11.1)		84.26 (12.6)	
Alcohol consumption (g/day)	Non-consumers	44,149	15.5	351	18.0
	>0–3	90,629	31.8	631	32.3
	>3–12	86,858	30.4	560	28.6
	>12–24	38,858	13.6	259	13.2
	>24	24,924	8.7	154	7.9
	mean (SD)	8.10 (11.7)		7.53 (11.1)	
Smoke status	Never	156,085	54.7	1194	61.1
	Former	66,275	23.2	422	21.6
	Smoker	56,531	19.2	298	15.2
	Unknown	6527	2.3	41	2.1
Physical activity	Inactive	154,608	54.2	1152	58.9
	Active	125,497	44.0	767	39.2
	Unknown	5313	1.9	36	1.8
Age at menarche (years)	<12	41,017	14.4	320	16.4
	12	58,727	20.6	417	21.3
	13	71,924	25.2	452	23.1
	>13	103,538	36.6	703	36.0
	Unknown	10,212	3.6	63	3.2
Menopausal status	Perimenopause	52,144	18.3	389	19.9
	Premenopause	108,603	38.1	414	21.2
	Postmenopause	124,671	43.7	1152	58.9
Age at menopause (years)	<45	9675	3.4	47	2.4
	45 to 50	29,226	10.2	183	9.4
	50 to 55	44,410	15.6	477	24.4
	≥55	8427	3.0	151	7.7
	Unknown	193,680	67.9	1097	56.1
Standard Menstrual Cycle (years)	<20	46,227	16.2	74	3.8
	20 to 30	70,761	24.8	293	15.0
	30 to 40	93,599	32.8	946	48.4
	>40	8236	2.9	173	8.8
	Unknown	66,595	23.3	469	24.0
Number of live births	0	41,971	14.7	311	15.9
	1	42,260	14.8	309	15.8
	2	108,384	38.0	733	37.5
	3	50,681	17.8	355	18.2
	4 or more	20,836	7.3	132	6.8
	Unknown	21,286	7.5	115	5.9
Ever use of hormonal treatment for menopause	No	200,814	70.4	1203	61.5
	Yes	64,045	22.4	610	31.1
	Unknown	20,559	7.2	142	7.3
Ever use of contraceptive pill	No	104,972	36.8	1091	55.8
	Yes	172,250	60.4	815	41.7
	Unknown	8196	2.9	49	2.5
Diabetes Mellitus	Yes	5327	1.9	68	3.5
	No	258,034	90.4	1658	84.8
	Don’t know	916	0.3	12	0.6
	Unknown	21,141	7.4	217	11.1

Except for values where the mean and standard deviation (SD) are specified, all values are presented as the total number (N) and %.

**Table 2 nutrients-17-01645-t002:** Multivariable hazard ratios (HR) and 95% confidence intervals (CI) of adherence to Empirical Dietary Index for Insulin Resistance (EDIR), Empirical Dietary Index for Hyperinsulinemia (EDIH), Diabetes Risk Reduction Diet (DRRD), and EC risk among the EPIC population.

Dietary Patterns	Models	T1	T2	T3	P_trend_
		HR (95% CI)	HR (95% CI)	HR (95% CI)	
EDIR	n (events)	95,140 (568)	95,139 (661)	95,139 (726)	
	Model 1	1.00 (Reference)	1.05 (0.93 to 1.18)	1.17 (1.04 to 1.31)	0.008
	Model 2	1.00 (Reference)	0.98 (0.87 to 1.10)	1.03 (0.91 to 1.16)	0.61
EDIH	n (events)	95,140 (729)	95,139 (728)	95,139 (498)	
	Model 1	1.00 (Reference)	1.03 (0.92 to 1.16)	1.12 (0.99 to 1.26)	0.07
	Model 2	1.00 (Reference)	0.97 (0.86 to 1.09)	1.00 (0.88 to 1.12)	0.95
DRRD	n (events)	102,497 (593)	109,437 (660)	73,484 (702)	
	Model 1	1.00 (Reference)	0.95 (0.85 to 1.06)	0.96 (0.85 to 1.09)	0.51
	Model 2	1.00 (Reference)	0.98 (0.88 to 1.09)	1.02 (0.91 to 1.16)	0.76

Model 1: Multivariable model stratified by age at recruitment and country, and adjusted by menopausal status (premenopause, postmenopause), smoking status (never, former, active smoker) and ever use of hormone treatment for menopause (yes, no). Model 2: Multivariable model stratified by age at recruitment and country, and adjusted by menopausal status, smoking status, ever use of hormone treatment for menopause and BMI (kg/m^2^ continuous). Tertil 1: For EDIH and EDIR 95,140 participants and for DRRD 102,497. Tertil 2: For EDIH and EDIR 95,139 participants and for DRRD 109,437. Tertil 3: For EDIH and EDIR 95,139 participants and for DRRD 73,484.

**Table 3 nutrients-17-01645-t003:** Multivariable HR and 95% CI of overall and endometrial cancer-specific mortality according to the adherence to DRRD, EDIR, and EDIH in the EPIC population.

Mortality							
		T1		T2		T3	P_trend_
	n (Deaths)	HR (95% CI)	n (Deaths)	HR (95% CI)	n (Deaths)	HR (95% CI)	
EDIR	652 (130)	1.00 (Reference)	651 (116)	0.83 (0.60 to 1.16)	652 (134)	1.03 (0.74 to 1.42)	0.95
EDIH	652 (123)	1.00 (Reference)	651 (131)	1.27 (0.92 to 1.73)	652 (126)	0.99 (0.70 to 1.41)	0.88
DRRD	729 (154)	1.00 (Reference)	728 (133)	0.98 (0.71 to 1.34)	498 (93)	0.87 (0.62 to 1.22)	0.42
**Endometrial Cancer Specific Mortality**							
		**T1**		**T2**		**T3**	**P_trend_**
	**n (Deaths)**	**HR (95% CI)**	**n (Deaths)**	**HR (95% CI)**	**n (Deaths)**	**HR (95% CI)**	
EDIR	652 (44)	1.00 (Reference)	651 (42)	0.86 (0.51 to 1.45)	652 (47)	0.99 (0.58 to 1.68)	0.92
EDIH	652 (47)	1.00 (Reference)	651 (47)	0.94 (0.57 to 1.54)	652 (39)	0.74 (0.42 to 1.30)	0.31
DRRD	729 (49)	1.00 (Reference)	728 (48)	0.91 (0.54 to 1.54)	498 (36)	0.90 (0.53 to 1.55)	0.72

## Data Availability

The data presented in this study is preserved by the EPIC centers. Data are available for investigators who seek to answer important questions on health and disease in the context of research projects that are consistent with the legal and ethical standard practices of IARC/WHO and the EPIC centers. The primary responsibility for accessing the data belongs to IARC and the EPIC centers. Access to materials from the EPIC study can be requested by contacting epic@iarc.fr.

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
