# Peer review of "Diabetes-Related Dietary Patterns and Endometrial Cancer Risk and Survival in the European Prospective Investigation into Cancer and Nutrition Study"

_nutrients, 2025, doi:10.3390/nu17101645_

Round 1
Reviewer 1 Report
Comments and Suggestions for Authors
This manuscript presents a prospective analysis from the EPIC cohort, investigating the associations between three diabetes-related dietary patterns and the risk and mortality of endometrial cancer (EC). The following concerns should be resolved:
Major
The attenuation of the association between EDIR and EC risk after adjusting for BMI is a pivotal finding. However, the manuscript interprets this attenuation as evidence that BMI "explains" the relationship. While plausible, this assumes that BMI is a mediator, not a confounder — a distinction that remains unaddressed.Consider discussing the potential mediation role of BMI (e.g., as part of the causal pathway from diet to EC) versus its role as a confounder. Including a formal mediation analysis (e.g., using marginal structural models or mediation decomposition) could strengthen causal inference.
The lack of concordance between EDIR and EDIH (despite high correlation, r=0.85) deserves further exploration. Is EDIR a better proxy for chronic metabolic dysfunction relevant to EC than EDIH? Could this relate to different underlying biological pathways?
Discuss whether DRRD’s broader nutritional scope (beyond insulin resistance) may dilute associations with EC. The manuscript could consider whether other diet scores more tightly linked to inflammation or estrogen metabolism might be more predictive.
Given the strong interaction between hormonal milieu and EC risk, it would be valuable to more explicitly discuss whether the effect of dietary patterns differs pre- vs post-menopause. Including p-values for interaction in the subgroup tables would help assess statistical heterogeneity.
Consider briefly highlighting how insulin resistance might amplify estrogen signaling in adipose-rich postmenopausal women, contributing to EC development — possibly by downregulating SHBG, leading to increased bioavailable estradiol.
The dietary data were collected at a single time point at baseline, which may not reflect long-term dietary habits. Although acknowledged in the limitations, this issue deserves more emphasis in interpreting null findings for mortality.
The absence of associations between dietary patterns and EC-specific mortality might reflect limited power (only 133 EC-specific deaths). Consider discussing the implications of statistical power limitations on these findings.
Minor
Ensure consistent use of abbreviations (e.g., EC, BMI, DRRD) throughout the text and tables.
Reviewer 2 Report
Comments and Suggestions for Authors
Dear Authors,
please provide additional explanations for the following issues:
1. Please present in the methodology how the patients were prepared for anthropometric measurements. Were the measurements performed independently by the subjects or by qualified personnel?
2. Please provide a detailed description of the equipment used for anthropometric measurements and the number of measurements performed.
3. Were the same measurement procedures followed in all centers and was the same equipment used?
4. Please include information on the FFQ method in the methodology, i.e. the number of products included and presentation of all categories of frequency of consumption. Was the FFQ used validated?
5. How did the subjects complete the questionnaires - online or in person? Was it possible to clarify any doubts while completing the questionnaire?
6. Were the subjects diagnosed by a doctor for diabetes, or was this information provided by the subjects?
7. Was physical activity assessed based on self-assessment by the study participants or was it assessed using another assessment method?
Kind regards
Round 2
Reviewer 1 Report
Comments and Suggestions for Authors
My concerns have been addressed.